# Immune Response Related to Lymphadenopathy Post COVID-19 Vaccination

**DOI:** 10.3390/vaccines11030696

**Published:** 2023-03-17

**Authors:** Tzu-Chuan Ho, Daniel Hueng-Yuan Shen, Chin-Chuan Chang, Hung-Pin Chan, Kuo-Pin Chuang, Cheng-Hui Yuan, Ciao-Ning Chen, Ming-Hui Yang, Yu-Chang Tyan

**Affiliations:** 1Department of Medical Imaging and Radiological Sciences, Kaohsiung Medical University, Kaohsiung 807, Taiwan; 2Department of Nuclear Medicine, Kaohsiung Veterans General Hospital, Kaohsiung 813, Taiwan; 3Department of Nuclear Medicine, Kaohsiung Medical University Hospital, Kaohsiung 807, Taiwan; 4School of Medicine, Kaohsiung Medical University, Kaohsiung 807, Taiwan; 5Neuroscience Research Center, Kaohsiung Medical University, Kaohsiung 807, Taiwan; 6Department of Electrical Engineering, I-Shou University, Kaohsiung 840, Taiwan; 7Graduate Institute of Animal Vaccine Technology, College of Veterinary Medicine, National Pingtung University of Science and Technology, Pingtung 912, Taiwan; 8Mass Spectrometry Laboratory, Department of Chemistry, National University of Singapore, Singapore 119077, Singapore; 9Department of Medical Education and Research, Kaohsiung Veterans General Hospital, Kaohsiung 813, Taiwan; 10Center of General Education, Shu-Zen Junior College of Medicine and Management, Kaohsiung 821, Taiwan; 11Research Center for Precision Environmental Medicine, Kaohsiung Medical University, Kaohsiung 807, Taiwan; 12Department of Medical Research, Kaohsiung Medical University Hospital, Kaohsiung 807, Taiwan; 13Center for Tropical Medicine and Infectious Disease Research, Kaohsiung Medical University, Kaohsiung 807, Taiwan

**Keywords:** COVID-19, C19-VAL, lymphadenopathy, lymph node

## Abstract

Mass vaccination against coronavirus disease 2019 (COVID-19) is a global health strategy to control the COVID-19 pandemic. With the increasing number of vaccinations, COVID-19 vaccine-associated lymphadenopathy (C19-VAL) has been frequently reported. Current findings emphasize the characteristics of C19-VAL. The mechanism of C19-VAL is complicated to explore. Accumulated reports separately show that C19-VAL incidence is associated with receiver age and gender, reactive change within lymph nodes (LN), etc. We constructed a systematic review to evaluate the associated elements of C19-VAL and provide the mechanism of C19-VAL. Articles were searched from PubMed, Web of Science and EMBASE by using the processing of PRISMA. The search terms included combinations of the COVID-19 vaccine, COVID-19 vaccination and lymphadenopathy. Finally, sixty-two articles have been included in this study. Our results show that days post-vaccination and B cell germinal center response are negatively correlated with C19-VAL incidence. The reactive change within LN is highly related to C19-VAL development. The study results suggested that strong vaccine immune response may contribute to the C19-VAL development and perhaps through the B cell germinal center response post vaccination. From the perspective of imaging interpretation, it is important to carefully distinguish reactive lymph nodes from metastatic lymph node enlargement through medical history collection or evaluation, especially in patients with underlying malignancy.

## 1. Introduction

In December 2019, a novel coronavirus causing severe acute respiratory syndrome coronavirus 2 (SARS-CoV-2) was first discovered in the city of Wuhan, China. SARS-CoV-2 infected individuals can develop coronavirus disease 2019 (COVID-19), which ranges from mild to fatal symptoms. Up to November 2022, the COVID-19 pandemic caused 641.47 million infected cases and 6.63 million deaths in the entire world [1]. The global vaccination campaign began in December 2020, with the goal to end the COVID-19 pandemic, in which the mRNA vaccines (Pfizer/BioNTech, New York, NY, USA, BNT162b2 and Moderna, Cambridge, MA, USA, mRNA-1273) and viral recombinant vaccine (AstraZeneca, Cambridge, UK, ChAdOx1) are widely used (and a few inactive vaccines were also used) [1]. Although side effects of vaccines are continued to be reported, vaccination is still recommended, since its benefit outweighs the risks of adverse effects [2]. With the administration of vaccines, rare adverse events post COVID-19 vaccinations have possibly increased.

In the clinical trials of COVID-19 vaccination, painful or nonpainful swelling or enlargement of the lymph node is an occasional side effect of vaccination (≤1.1% of incidence). These symptoms belong to COVID-19 vaccine-associated lymphadenopathy (C19-VAL) [3,4,5]. Recently, C19-VAL has been frequently discovered in healthy individuals in the setting of routine mammography [6,7] as well as in cancer patients who are under cancer staging or follow-up [7,8,9]. The C19-VAL incidence in patients who undergo CT scans [10], FDG-PET/CT [11,12,13], breast MRI [14] and ultrasound [15,16] is around 9 to 54%. According to the studies regarding imaging follow up, C19-VAL is a self-limiting disease [17,18,19]. The duration of C19-VAL is up to 10 days and resolved in two months [13,20]. The site of C19-VAL is usually found in the axillary region, then next in the supraclavicular and cervical regions [17,19,21,22,23,24], but which also occurs in malignancy lymphadenopathy. In addition, the features of lymph nodes (such as round, hilum absence, symmetrical cortex) in the patients with C19-VAL mimic that in the patients with the malignancy lymphadenopathy [18,25,26]. However, the studies regarding to C19-VAL focused on characterizing the feature and duration of C19-VAL. The mechanism of C19-VAL is yet to be explored.

VAL has been reported as a rare finding in vaccine administration for human papillomavirus [27,28], influenza [29,30], smallpox [31], measles [32] and tuberculosis [33]. It is due to reactive changes within the lymph node that are separately well documented in vaccinations for human papillomavirus [27] and smallpox [34]. The reactive change was also discovered in the lymph node from C19-VAL, displaying its involvement in the cause of C19-VAL [8,35,36,37]. In addition, other reports show that C19-VAL incidence is associated with age and gender [13,38], even occurring in other adverse vaccinal effects such as Kikuchi-Fujimoto disease [39,40] and autoimmune disease [41]. In this study, we aimed to construct a systematic review that not only analyzed the associated effects in the incidence of C19-VAL, but also explored the mechanism of C19-VAL.

## 2. Materials and Methods

The guidelines of PRISMA are shown in Figure 1. Articles were searched from the database including PubMed, Web of Science and EMBASE from 1 December 2020 to 31 October 2022. The combinations of terms included COVID-19 vaccine, COVID-19 vaccination and lymphadenopathy, which were used to search articles (Appendix A). Duplicate records were removed through Excel screening. This review focuses on clinical findings regarding associated elements of C19-VAL. We only included the patients with sudden swollen or enlargement lymph node post COVID-19 vaccination, regardless of medical history prior or post vaccination. In this study, not only full research articles, but also single patient case reports were included. The irrelevant articles were also removed by title and abstract screening.

## 3. Results

### 3.1. Studies including and Characterization

The results of the article search and selection were summarized in the PRISMA flowchart (Figure 1). A total of 2057 records were identified from the database, including PubMed, Web of Science and EMBASE. There were 1294 studies removed due to duplicates at the initial screening, and then 540 articles were excluded by title and abstract at stage two screening. The remained 223 studies for screening. A total of 161 relevant references were further assessed but were eliminated if they only included review, conference abstract, commentary, full-text unavailable, not written in English or without associated elements of C19-VAL. Finally, only 62 articles were included in this systematic review and are characterized in Table 1. 

These studies were conducted in 17 countries (France: 1; New Zealand: 1; Portugal: 1; Belgium: 2; Canada: 2; China: 2; Singapore: 2; Taiwan: 2; Germany: 2; Israel: 1; Switzerland: 3; Spain: 4; Italy: 4; Korea: 5; UK: 6; Japan: 9; USA: 15). Types of articles contained single center retrospective studies (n = 10), case series (n = 6) and case reports (n = 46). A total of 14 elements were described which are associated with the C19-VAL occurrence. Those elements included site, type, dosage, days post-vaccination, age, gender, reactive change, cancer, inflammation, Kikuchi-Fujimoto disease (KD), immunosuppression, pathogens, autoimmune disease or idiopathic multicentric Castleman disease (iMDC), of which reactive change was mostly reported (Figure 2).

### 3.2. Negative Correlation between Days Post-Vaccination and C19-VAL Incidence

The site, type, dosage and days post-vaccination for C19-VAL incidence was individually evaluated in the eight reports, which included six single-center retrospective studies, one case series and one case report (Table 1, [10,13,18,19,20,42,43,44]). The case series study, which recruited 20 females with C19-VAL in supraclavicular LN, addresses the association between the injection site of the vaccine and C19-VAL incidence [19]. This study indicated that 85% (17/20) of participants with a high injection site had C19-VAL. However, this phenomenon was based on the voluntary feedbacks, and the control group was missing. The effects of the high injection site in C19-VAL occurrence should be further evaluated. Two single-center retrospective studies from the UK and Korea individually evaluated the correlation between the vaccine types and C19-VAL incidence [13,20]. The study in UK recruited 204 participants with malignancy (mean aged 68 ± 11 years) and 51% of participants were vaccinated with AstraZeneca (n = 43) and Pfizer-BioNTech (n = 62). The C19-VAL incidence in AstraZeneca and Pfizer-BioNTech was 48.83 and 25.8%, respectively. This study indicated that the vector vaccine can increase the C19-VAL incidence in cancer patients [13]. The study in Korean analyzed the C19-VAL incidence in healthy participants (mean aged 48 ± 12 years) who received mRNA vaccines (n = 349) and AstraZeneca (n = 64). The C19 VAL incidence in AstraZeneca was 7.8%; significantly lower than that of mRNA vaccines (56%). This study showed that C19-VAL incidence was significantly higher in the healthy population who received the mRNA vaccine [20]. The demographics of these two studies were different in the age and medical history of participants. In addition, there were no other studies to support the individual finding. It is still a question of the influence of vaccine type in the C19-VAL incidence. Similarly, the relevance between vaccine dosage and C19-VAL incidence was not consistent in the four single center retrospective studies due to different demographics such as age and medical history [18,20,42,43]. Most findings indicated that the first dose positively correlates with C19-VAL incidence when healthy and cancer groups were analyzed together [18,20,42]. The linking of days post-vaccination to C19-VAL incidence was addressed in three single center studies from Korea [20], Israel [43] and the US [44]. The study in Korea (n= 413) indicated that C19-VAL incidence in the healthy group was decreased in the following days post-vaccination (incidence of 38% in D1–14, 41% in D15–28 and 21% over D28) [20]. The similar finding was also shown in healthy individuals (n = 2304) in the study from the US (incidence of 2.3% in D1–14, 1.8% in D15–28 and 0.2% over D28) [44]. Contrarily, the study in Israel (n = 75) indicated that C19-VAL incidence in the cancer group was positively correlated with days post-vaccination (OR, 1.53; 95% CI, 1.18–1.99, *p*  =  0.005). The effects of days post-vaccination in C19-VAL incidence in the healthy groups were contrary to the cancer groups [20,43,44]. According to the two studies from Korea [20] and the US [44] (including more than 400 individuals), there is a negative correlation between the days post-vaccination and C19-VAL incidence for the healthy groups. 

### 3.3. Reactive Change and B Cell Germinal Center Related to the C19-VAL Development

In the 62 reports listed in Table 1, there were 53 articles which separately described the role of age, gender, reactive change, inflammation, KD and immunosuppression in C19-VAL (Table 1). The study number regarding each element is shown in Figure 2. Reactive change was most frequently reported. A total of 36 studies including one single-center retrospective study [45], five case series [19,46,47,48,49] and 30 case reports [8,50,51,52,53,54,55,56,57,58,59,60,61,62,63,64,65,66,67,68,69,70,71,72,73,74,75,76,77,78] revealed that C19-VAL is due to a reactive change in LN. It is an immune response for other vaccines [27,34]. Pathological characterization shown in the 14 of 36 studies, including two case series [47,49] and 12 case reports [8,59,62,65,68,69,71,72,73,74,76,78], which directly shows the characterization of reactive change, such as reactive lymphadenitis, reactive hyperplasia, reactive follicular hyperplasia, extrafollicular proliferation of B-blasts and small lymphoid cells. Due to a lack of proper controls in those studies, current findings only indicated that reactive change is highly correlated with C19-VAL. 

Several reports of case series and case reports indicated the association of inflammation and KD with C19-VAL development [24,36,38,39,40,79,80,81,82,83]. Some of the studies indicated that the inflammation of C19-VAL shows the abundant inflammatory cells in the biopsy [24,36] or high choline uptake in the imaging of LN [79]. Four case reports found C19-VAL with necrotizing lymphadenitis in LN biopsy, which were diagnosed as KD [39,40,81,82]. Those findings were based on studies in small populations without control groups. The real correlation between C19-VAL incidence and inflammation or KD is not clear. Five studies (one single-center retrospective study and four case reports) indicated the effect of cancer for C19-VAL [43,84,85,86,87]. The single-center retrospective study indicated that the C19-VAL incidence was significantly lower in participants with hematological malignancy (incidence, 32%, 24/75) than those of non-hematological malignancy (incidence, 53.6%, 183/341) [43]. Four case reports, involving one to two participants per study, showed that persistent lymphadenopathy occurred after vaccination. Patients were then diagnosed as marginal zone B-cell lymphoma (MZL) (n = 1) [84], large B-cell lymphoma (DLBCL) (n = 2) [85], angioimmunoblastic T cell Lymphoma (AITL) (n = 1) [86] or lymphangioma (n = 1) [87]. These findings suspected that vaccination may induce the hematological malignancy. However, these are uncommon symptoms from a small number of reports. The relationship between hematological malignancy and C19-VAL is uncertain. Since there was no other similar report, it is difficult to define the correlation between hematological malignancy and C19-VAL incidence.

Consistent findings were observed in the roles of age, gender and immunosuppression in the C19-VAL incidence. A total of four single-center retrospective studies from Japan [38], Korea [20], the UK [13] and Israel [43] revealed the C19-VAL incidence is significantly higher in the younger age group. Two of these studies also showed that similar findings in the female group [13,38] were consistent with two other retrospective studies [10,45]. The single-center retrospective study from Israel showed that C19-VAL incidence was significantly lower in patients with immunosuppressive treatment (incidence, 30%, 25/82 and 45.09%, 170/377 without treatment) [43]. Another single-center retrospective study from Germany further found that the C19-VAL incidence in recent anti-CD20 treatment for individuals with hematological malignancy is significantly lower than that without recent anti-CD20 treatment (8.8 and 41.4%, respectively) [88]. These findings directly indicated that B cell germinal center responses are positively correlated with C19-VAL incidence. Four case reports showed that pathogen reactivation (TB (n = 1) [66] and EBV(n = 1) [89]), iMDC (n = 1) [90] and autoimmune disease (ITP) (n = 1) [41] are correlated with C19-VAL development. Those findings indicated that reactive change is also related to C19-VAL development. The pathological findings revealed that immune cell activation and proliferation are involved in the reactive change within LN [8,47,49,59,62,65,68,69,71,72,73,74,76,78]. Of these, the B cell germinal center response plays important roles in C19-VAL incidence based on results with anti-CD20 treatment [88].

**Table 1 vaccines-11-00696-t001:** Clinical demographics in study for associated elements in the C19-VAL.

Design	Study	Country	Participants	Age (Years)	Medical History	Male N (%)	Vaccine Type	Vaccine Dose	Last Vaccine to C19-VAL (Days)/Site	Main Finding	Elements
Single center reprospective study	Yoshikawa T. (2022) [38]	Japan	433	65 ± 11	No past and current LAD related disease and COVID-19	300 (69.28)	NR	2 (most)	NR/all for ipsilateral axillary	Incidence of C19-VAL is significantly higher in young age and female	Young age and female
	Shin M. (2021) [45]	Korea	31	45 ± 5	No history of malignancy, vaccination before 18F-FDG PET/CT	11 (35)	AstraZeneca	NR	4–29/bilateral axillary, supraclavicular	Percentage of C19-VAL is significantly higher in female, FDG-avid deltoid muscle can be a helpful sign to presume the reactive LN	Female and reactive change
	Park JY. (2022) [20]	Korea	413	48 ± 12	No history of malignancy, vaccination within 12 weeks, vaccination before ultrasonography	10 (2.42)	Moderna (19); AstraZeneca (64); Pfizer-BioNTech (330)	2 (257)	1–82/axillary	48.9% cases with C19-VAL. Incidence of C19-VAL is significantly higher in young age, mRNA vaccine and post 1st dose as well as decreased as days from vaccination	Young age, mRNA vaccine, 1st dose, and days post-vaccination
	Nishino M. (2021) [10]	USA	232	40–96	All with lung cancer, CT scans prior and post vaccination	88 (37.9)	Moderna (28); Pfizer-BioNTech (204)	2	7–68/axillary, subpectoral	9% cases with C19-VAL. Incidence of C19-VAL is significantly higher in female and in Moderna	Female and Moderna
	Cocco G. (2021) [18]	Italy	24	25–74	Without fever and no history of hematological malignancy, autoimmune disease and vaccination before ultrasonography	10 (41.6)	Moderna (3); AstraZeneca (8); Pfizer-BioNTech (13)	Least 1	NR/axillary, supraclavicular	All cases with C19-VAL Percentage of C19-VAL is significantly higher post 1 dose vaccination	1st dose
	El-Sayed MS. (2021) [13]	UK	204	68 ± 11	Without LAD pathologies, vaccination within 12 weeks and before ^18^F-FDG PET/CT	98 (48)	AstraZeneca (43); Pfizer-BioNTech (62); unknown (99)	2 (87)	Up to 70/axillary	36% cases with C19-VAL. Incidence of C19-VAL is significantly higher in young age, vector vaccine and female	Young aged, vector vaccine and female
	Ah-Thiane L. (2022) [42]	France	226	67–76	Most prostate cancer, vaccination before ultrasonography, MRI, or ^18^F-FDG PET/CT	212 (93.8)	Moderna (11); AstraZeneca (60); Pfizer-BioNTech (152); Janssen (3)	2 (124)	14–51/axillary and supraclavicular	42.5% cases with C19-VAL. Incidence of C19-VAL was significant higher post the 1st vaccination	1st dose
	Maimone S. (2022) [44]	US	2304	30–92	Vaccination before screening mammography	NR	Moderna (1109); Pfizer-BioNTech (1135); other (41); unknown (18)	2 (1883)	0–28 and >28/ipsilateral axillary	1% cases with C19-VAL. Incidence of C19-VAL was significantly decreased as days from vaccination	Days post-vaccination
	Eifer M. (2022) [43]	Israel	426 Immunosuppressive treatment (82), hemato-logical malignancy (75)	67 ± 12	Vaccination before PET/CT and without malignancy involving axillary LN	219 (51)	Pfizer-BioNTech	2 (103)	5–18/axillary	Incidence of C19-VAL is significantly higher in young age, 2nd vaccination and increased in days post last vaccination, but lower in immunosuppressive treatment and hematologic malignancy	Young age, 2nd vaccination, days post-vaccination, immunosuppressive, and hematologic malignancy
	Cohen D. (2021) [88]	Germany	137 Recent anti-CD20 (34), no recent anti-CD20 (68)	>16	Hematologic malignancy without MHL	75 (54.7)	NR	Either of 1 or 2	6–27/ipsilateral axillary or supraclavicular	Incidence of C19-VAL in recent anti-CD20 vs. no recent anti-CD20 (8.8 vs. 41.1%, significant)	B cell germinal center response
Case series	Fernández-Prada M. (2021) [19]	Spain	20	20–60	Autoimmune disease (4), thyroid cancer (2)	0	Pfizer-BioNTech (19) Moderna (1)	2 (14)	0–4/supraclavicular	12 of 20 patients reported high injection site, biopsy of LN from two type vaccines revealed reactive change	High injected site and reactive change
	García-Molina F. (2021) [36]	Spain	6	27–62	NR	NR	Pfizer-BioNTech	1	5/axillary or supraclavicular	Cytological for FNA and biopsy: nonspecific chronic adenitis, resolution of ALAD and SLAD after anti-inflammation drug	Inflammation
	Özütemiz C. (2021) [48]	USA	2	62	Current metastatic breast cancer (1), two cancer history	1 (50%)	Moderna (1) Pfizer-BioNTech (1)	3	1–2/ipsilateral axillary	Due to imaging and vaccine history, reactive LN	Reactive change
	Heaven CL. (2022) [47]	New Zealand	5	41–76	High suspicion of cancer	2 (40%)	Pfizer-BioNTech	2	7–34/bilateral cervical	Biopsy: reactive follicular hyperplasia with no evidence of atypia or malignancy	Reactive change
	Hagen C. (2021) [49]	Switzerland	5	41–66	Lung cancer (2), neuroendocrine tumor (1)	2 (40%)	Pfizer-BioNTech (2) Moderna (3)	1 (3)	3–33/bilateral axillary, supraclavicular	FNA: reactive follicular hyperplasia	Reactive change
	Brown AH. (2021) [46]	UK	2	48, 67	Breast cancer for right (1) and for left (1)	0	NR	NR	14, 21/ipsilateral axillary, subpectoral	FNA: reactive change without malignancy	Reactive change
Case report	Goldman S. (2021) [86]	Belgium	1	66	Hypercholesterolemia, type 2 diabetes, recent cervical lymphadenopathies	1	Pfizer-BioNTech	2	150/ipsilateral supraclavicular, cervical, left axillary and abdomen	FNA of CLN: atypical T cell infiltrate with high endothelial venules proliferation; NGS: positive AITL, the number and distribution of LAD increased after 3rd vaccination	AITL
	Wolfson S. (2022) [50]	USA	2	50, 60	No (1), simultaneously left ductal carcinoma	0	Moderna	1	10, 63/ipsilateral axillary	FNA: begin reactive LN for no medical history, biopsy: metastatic adenocarcinoma for left ductal carcinoma patients	Reactive change
	Mizutani M. (2022) [85]	Japan	2	67, 80	NR	1	Pfizer-BioNTech	2	14, 1/left axillary	Persistent ALAD from 1st dose vaccination and gradually enlarged post 2nd dose, finally diagnosed as DLBC by IHC of biopsy	DLBCL
	Sekizawa A. (2022) [84]	Japan	1	80	Hypertension, angina pectoris, mitral valve regurgitation, ovarian tumor	0	Pfizer-BioNTech	2	21/ipsilateral temporal cervical, submandibular, and jugular	Persistent ipsilateral temporal after 1st dose vaccination, and sudden enlarged post 2nd dose, finally diagnosed as MZL	MZL
	Sasa S. (2022) [87]	Japan	1	80	Right breast	0	Pfizer-BioNTech	2	90/ipsilateral axillary	Multilobulated cystic mass and branches on ultrasonography, finally diagnosed as lymphangioma and resected	Lymphangioma
	Saito K. (2022) [41]	Japan	1	66	Current malaise and oral bleeding and purpura	0	Pfizer-BioNTech	1	2/systematic	Low platelet count, markedly increased megakaryocytes in bone marrow, and present of serum anti-glycoprotein IIb/IIIa, finally diagnosed ITP	ITP
	Hoffmann C. (2022) [90]	Germany	1	20	Fever, centigrade, loss of appetite, malaise, weakness, and exertional dyspnea post vaccination	1	Pfizer-BioNTech	2	18/supraclavicular, axillary	iMDC post vaccination	iMDC
	Girardin FR. (2022) [89]	Switzerland	1	40	Recent EBV infection	0	Moderna	2	1/bilateral axillary and supraclavicular	EBV positive parafollicular immunoblastic cells in LN, induce repeated and extend C19-VAL through enhancing the vaccine immunity	EBV
	Cha HG. (2022) [66]	Korea	1	66	Injection site tenderness and fatigue post 1st dose, acute idiopathic thrombocytopenic purpura	0	AstraZeneca	2	3/ipsilateral supraclavicular	Persistent LAD up to 8 weeks, further revealed multiple FDG avid-LNs not limited in supraclavicular, LN biopsy of SLN: chronic granulomatous inflammation, PCR positive for TB	TB
	Ganga K. (2021) [24]	USA	1	58	Hypertension	1	Moderna	NR	2/left neck	FNA for LN of left neck: negative for malignancy and positive for inflammatory cells, improvement and resolution of neck swelling and dysphagia by antibiotic treatment	Inflammation
	Cheong KM. (2022) [80]	Taiwan	1	32	NR	0	AstraZeneca	1	2/lower neck	Neck lymphadenitis diagnosed on ultrasonography	Inflammation
	Andresciani F. (2022) [79]	Italy	1	62	Prostate cancer	1	Pfizer-BioNTech	2	21/ipsilateral axillary, paratracheal, paraaortic, subcarinal, and bilateral hilar	Choline intensity decreased in LN, finally diagnosed as inflammatory LN, not oncological disease	Inflammation
	Tsumura Y. (2022) [83]	Japan	1	31	Metastatic Ewing sarcoma	0	Pfizer-BioNTech	NR	21/ipsilateral axillary	An inflammatory lesion rather than metastatic lymph node swelling	Inflammation
	Tan HM. (2021) [39]	Singapore	2	18, 34	Current fever	1	Pfizer-BioNTech	1–2	17–35/left axillary, supraclavicular, subpectoral	Fever, transient leukopenia, LAD, negative for infection, necrotizing lymphadenitis in LN biopsy, finally diagnosed KD	KD
	Caocci G. (2022) [40]	Italy	1	38	Recent fever for ten day, chills, and fatigue, C19-VAL post 1st dose	0	Pfizer-BioNTech	2	31/left axillary	Fever, negative for infection, leukopenia, LAD, and biopsy of LN: histiocytic necrotizing lymphadenitis (numerous CD68+ histiocytes and CD3+ T cells, few CD20+ B cells), finally diagnosed KD	KD
	Kashiwada T. (2022) [82]	Japan	1	27	Recent repeated fever, C19-VAL post 1st dose	0	Pfizer-BioNTech	2	68/ipsilateral axillary	Fever, negative for infection, leukopenia, LAD, and necrotizing lymphadenitis in LN biopsy, finally diagnosed KD	KD
	Guan YY. (2022) [81]	China	1	36	Current fever and fatigue		Sinopharm	1	28/left cervical, neck	Fever, LAD and necrotizing lymphadenitis (numerous CD68+ histiocytes and CD3+ T cells, few CD20+ B cells) in LN, finally diagnosed KD	KD
	Xu GY. (2021) [51]	USA	1	72	Mantle cell lymphoma	1	Pfizer-BioNTech	NR	2/ipsilateral axillary	With FDG-avid deltoid muscle, reactive LN, recurrent lymphoma	Reactive change
	Özütemiz C.(2021) [8]	USA	2	38, 46	Breast cancer (1)	0	Pfizer-BioNTech	2 (1)	8–15/ipsilateral axillary, supraclavicular	Biopsy: reactive follicular hyperplasia in lymph node without any evident of breast cancer and malignancy	Reactive change
	Nawwar AA. (2021) [52]	UK	1	75	Prostate cancer	1	AstraZeneca	1	3/ipsilateral axillary	With ^18^F-Choline-avid left deltoid muscle, reactive active LN	Reactive change
	Mitchell OR. (2021) [53]	UK	2	47, 55	NR	0	NR	NR	3/ipsilateral supraclavicular	Reactive LN by clinical and ultrasonographic examination	Reactive change
	Ulaner GA. (2021) [54]	Canada	1	68	Current right melanoma	1	Moderna	1	21/ipsilateral axillary	Unlike metastasis of right melanoma, reactive to vaccination	Reactive change
	Wong FC. (2022) [55]	USA	1	74	Prostate cancer	1	Moderna	2	6/ipsilateral axillary	Unlike metastasis of prostate cancer and findings for C19-VAL, considering to reactive to vaccination	Reactive change
	Garreffa E. (2021) [56]	UK	1	38	NR	0	Pfizer-BioNTech	1	7/ipsilateral clavicle	Reactive LN by ultrasonographic examination	Reactive change
	Prieto PA. (2021) [57]	USA	1	48	Melanoma	0	Moderna	1	5/ipsilateral axillary, neck	Biopsy: consistent with reactive LN and negative of melanoma	Reactive change
	Roca B. (2021) [58]	Spain	1	29	NR	0	Pfizer-BioNTech	1	7/ipsilateral supraclavicular	C19-VAL disappeared over the next few weeks	Reactive change
	Tan JHN. (2021) [59]	Singapore	1	34	No malignancy history	0	Pfizer-BioNTech	1	1/ipsilateral supraclavicular	FNA: reactive follicular hyperplasia	Reactive change
	Gable AD. (2021) [60]	USA	1	24	Current ED, never smoker, no medical or surgical history, no ED related disease	1	NR	2	4/ipsilateral axillary	ED due to typical bronchial carcinoid and LAD significant reduced later	Reactive change
	Suleman A. (2021) [61]	Canada	1	38	Current left Hodgkin lymphoma	0	Pfizer-BioNTech	1	7/ipsilateral axillary	Reduced later	Reactive change
	Tintle S. (2021) [62]	USA	1	23	Asthma, eczema, and hypothyroidism, simultaneously fever and acute kidney injury	0	Moderna	2	7/left axillary and abdomen	Biopsy of ALN: reactive lymphadenitis	Reactive change
	Weeks JK. (2021) [63]	USA	1	50	Current sigmoid adenocarcinoma	0	Moderna	2	30/bilateral axillary	Improvement of C19-VAL	Reactive change
	Mori M. (2022) [64]	Japan	1	30	NR	0	Pfizer-BioNTech	1	9/axillary	Resolution later	Reactive change
	Tzankov A. (2021) [65]	Switzerland	1	30	Current right papillary thyroid cancer	1	Moderna	1	21/left axillary	Biopsy: extrafollicular proliferation of B-blasts and resolution later	Reactive change
	Chan HP. (2022) [66]	Taiwan	1	71	Thyroid cancer, right renal cell carcinoma	1	Moderna	1	6/ipsilateral axillary	Due to imaging and vaccine history, reactive to vaccination	Reactive change
	Adin ME. (2022) [67]	USA	1	41	Simultaneously right breast cancer	0	Moderna	2	16/ipsilateral axillary,	Due to imaging and vaccine history, reactive to vaccination	Reactive change
	Kang ES. (2022) [68]	South Korea	1	59	Simultaneously SCC of the right mandibular gingiva	1	Moderna	2	10/ipsilateral axillary, bilateral cervical	FNA: only small lymphoid cells, reactive to vaccination	Reactive change
	Yu Q. (2022) [69]	China	1	34	Allergic disease, tuberculosis, past malignant tumors, recent infection, trauma	0	Sinovac	2	120/ipsilateral axillary	FNA: reactive hyperplasia and resolution later, reactive to vaccination	Reactive change
	Ashoor A. (2021) [70]	Italy	3	61–72	No (2), simultaneously breast cancer (1)	0	AstraZeneca	1 (1), 2 (2)	1–27/ipsilateral axillary	Imaging is normal and biopsy: benign reactive changes, reactive to vaccination	Reactive change
	Lee SM. (2022) [71]	South Korea	1	21	NR	1	Pfizer-BioNTech	2	2/ipsilateral axillary	Radial neuropathy associated with ipsilateral ALAD, FNA: reactive hyperplasia	Reactive change
	Kado S. (2022) [72]	Japan	1	31	NR	0	Pfizer-BioNTech	1	8/ipsilateral clavicle, scapular	FNA: follicular hyperplasia and resolution later, reactive to vaccination	Reactive change
	Aalberg JJ. (2021) [73]	USA	1	73	Metastatic renal cell carcinoma to lung and bone		Moderna	2	2/ipsilateral axillary	With FDG avid-ipsilateral deltoid muscle and FNA: polymorphous lymphoid population with no evidence of metastasis	Reactive change
	Cardoso F. (2021) [74]	Portugal	1	48	Usual contraceptive medication, Mercilon^®^	0	Pfizer-BioNTech	2	1/right cervical	Due to persistent LAD after first dose, FNA: reactive follicular hyperplasia	Reactive change
	Lam DL. (2022) [75]	USA	1	39	Simultaneously right breast cancer	0	Pfizer-BioNTech	2	1/ipsilateral axillary	C19-VAL resolution later, FNA for sentinel LN: negative metastasis and consistent with reactive to vaccination	Reactive change
	Musaddaq B. (2021) [76]	UK	1	57	Left breast cancer, simultaneously right breast cancer	0	Astra Zeneca	1	3/ipsilateral axillary	IHC: reactive LN with follicular hyperplasia and without metastasis cancer	Reactive change
	Dirven I. (2022) [77]	Belgium	1	60	MEN 1 syndrome and simultaneously right lung nodule	0	Pfizer-BioNTech	1	13/ipsilateral axillary	FNA: LN with a benign reactive pattern without metastatic disease	Reactive change
	Pudis M. (2021) [78]	Spain	1	30	Neuroendocrine tumor	0	Pfizer-BioNTech	2	40/bilateral axillary, unilateral supraclavicular and cervical	No infection, Biopsy: benign reactive LN with CD10+ B cell population, immune system activation to vaccination	Reactive change

NR: not reported; LN: lymph node fine needle aspiration; IHC: immunohistochemistry; ALAD: axillary lymphadenopathy; SLAD: supraclavivular lymphadenopathy; LAD: lymphadenopathy; MEN: multiple endocrine neoplasia; MHL: malignant axillary and supraclavicular hypermetabolic lymphadenopathy; ED: episode of hemoptysis; SCC: squamous cell carcinoma; AITL: angioImmunoblastic T cell Lymphoma ; NGS: next generation sequencing; iMDC: idiopathic multicentric castleman Disease; DLBL: large B-cell lymphoma; MZL: marginal zone B-cell lymphoma; HLH: hemophagocytic lymphohistiocytosis; KD: Kikuchi-Fujimoto disease; ITP: immune thrombocytopenia; TB: mycobaterium tuberculosis; EBV: Epstein–Barr virus.

## 4. Discussion

The study of side effects due to C19-VAL has been widely reported over the past two years. Several systematic reviews and meta-analyses have been published [90,91,92,93,94]. Those studies have investigated that the incidence [90,91,92,93], features [91,92], pathological findings [93], patient management [94] of C19-VAL. The object of this study is to evaluate the associated elements of C19-VAL and to provide the mechanism of C19-VAL. The study results found three elements are highly correlated with C19-VAL development or incidence. These elements are days post-vaccination, reactive change and B cell germinal center response. Due to the lack of control groups, current findings only suggest that reactive change is highly related to C19-VAL development. The remaining two elements (days post-vaccination and B cell germinal center response) have a negative effect on C19-VAL incidence.

Reactive change is a cause element for C19-VAL, which has been demonstrated in vaccinations for human papillomavirus [31] and smallpox [38]. Several pathological studies in this systematic review indicate that immune cell activation and proliferation are involved in the reactive change of LN [8,47,49,59,62,65,68,69,71,72,73,74,76,78]. Reactive change is a vaccine induced immune response [8]. The antigen processing and presentation between mRNA vaccine and adenovirus vector vaccine is different. The antigen-specific antibody and T cell responses of the mRNA vaccine were stronger than those of the adenovirus vector vaccine [95]. The fragmental mode of COVID-19 vaccine induces an immune response against SARS-CoV2 by stimulating adaptive immunity through cross-linking to the innate immune response of dendritic cells [96]. Initially, the locally activated antigen accumulates in the injection site. Subsequently, the antigen is processed by the dendritic cells through different innate receptors (TLR7 and MAD5 for mRNA vaccine; TLR9 for adenovirus vector vaccine). Then, activated dendritic cells migrate to regional lymph nodes. Lastly, dendritic cells present the antigen to the T cells, which then promote antibody production via the plasma B cells. As a result, large amounts of T cells and B cells exist in the LN. The lymphadenopathy immediately occurs in the LN after reactive hyperplasia (rapid extension) of T and B cells [97], which is a possible mechanism of C19-VAL. It also explains that the days of post-vaccination is a negative effect of C19-VAL incidence, due to the reduction of reactive hyperplasia in the following days post-vaccination. 

Anti-CD20 treatment is a B cell-depleting therapy to cure hematological malignancy. A previous study has demonstrated that patients who received anti-CD20 treatment present a low vaccine-induced protection against influenza A (H1N1) [98]. The cohort and observation studies showed that the anti-CD20 treatment can reduce the reactogenicity of mRNA vaccine [99,100,101] and AstraZeneca [101] resulting in low production of specific anti-SARS-CoV2 antibody. To our knowledge, the correlation between anti-CD20 treatment and C19-VAL is unclear. It was first found that B cell germinal center response is a negative element of C19-VAL incidence. B cell germinal center response in the lymph node contributes to the capability of cytotoxic T cells to kill infected cells and formation of antibody-secreting plasma B cells [102]. Recent studies show that the COVID-19 vaccines elicit a prominent B cell germinal center response in LN [103,104]. It reflects that there is a strong B cell germinal center response, which contributes to C19-VAL incidence.

Previous study results have also shown that the C19-VAL incidence is increased in younger ages and females. It was comparable with the previous study for female groups [10,105], which also found that this side effect was more common in women [106,107]. This might be because of results from the sex hormone-mediated immune response [108]. The association between aging and vaccine-induced protection shows that primary and secondary antibody responses to vaccination are impaired in the elderly [109]. Therefore, the strong vaccine-induced immune response may cause the C19-VAL. Recently, data from COVID-19 vaccination showed that the COVID-19 vaccine presents a high immunogenic response to induce T and B cell function [103,110]. There are several limitations in this systematic review. For example, the real correlation between reactive change and C19-VAL vaccination cannot be recognized because the control groups are missing. Additionally, those results may not reflect the fact that the B cell geminal center response also plays a negative role in C19-VAL in healthy receivers or patients with other solid tumors. Further investigation into the correlation between days post-vaccination and C19-VAL development for cancer patients is needed, due to small number of participants. KD is a necrotizing lymphadenitis characterized by lymphadenopathy and fever with leukopenia, thrombocytopenia and liver dysfunction [111]. As this is a small population study without a control group, the linkage of C19-VAL to KD needs to be assessed. Overall, those study results suggest that strong vaccine-induce immune response may contribute to C19-VAL development and incidence. This could be through the B cell germinal center response post vaccination. 

C19-VAL has caused a dilemma in diagnosis and patient management, especially in oncological patients [8,21]. It may lead to unnecessary biopsies and changes in therapy [8,108], which increase the psychological and medical burden of patients and the risk of those having inherent diseases. To avoid the above issues, current recommendations suggest that screening exams can be scheduled before the first dose or 4–12 weeks after the second dose of the COVID-19 vaccine [11]. For a booster dose, the exam should be provided at least 6 weeks post vaccination [112,113]. Based on the results of the literature review, it currently shows that C19-VAL may be caused by a strong vaccine-induced immune response. In the future, additional studies are needed with a better and more comprehensive study design to draw more tangible conclusions.

## 5. Conclusions

With mass COVID-19 vaccination, C19-VAL will be observed more frequently; more patients and clinicians will encounter medical dilemmas from C19-VAL. However, the vaccine benefits exceed the medical dilemmas of C19-VAL. C19-VAL is a common side effect and is recognized following the mass use of vaccines. The cause of C19-VAL is mostly considered as a reactive change for vaccination. The reactive change is an activation of immune cells by the vaccine to lymph nodes. Such effects can diminish with time post vaccination. It was further found that the B cell germinal center may contribute to C19-VAL. Although the B cell germinal center response may be a contributing factor to C19-VAL, global vaccination is one approach to end the COVID-19 epidemic. For now, it is not suggested to decline vaccination due to the cause of C19-VAL. From the perspectives of image interpretation, the positive finding of axillary C19-VAL reminds physicians that careful differentiation of reactive lymph nodes from metastatic lymphadenopathies via either the medical history and records or medical chart review, especially for the patients with underlying malignances, is important. Therefore, we suggest a follow-up imaging exam should be processed for the size and number of C19-VAL within two months of onset. If the C19-VAL is not resolved within this period, a biopsy examination should be suggested. In the future, additional studies are needed with a better and more comprehensive study design to draw the role of the B cell germinal center in C19-VAL. This will assist in reducing C19-VAL incidence post-vaccination.

## Figures and Tables

**Figure 1 vaccines-11-00696-f001:**
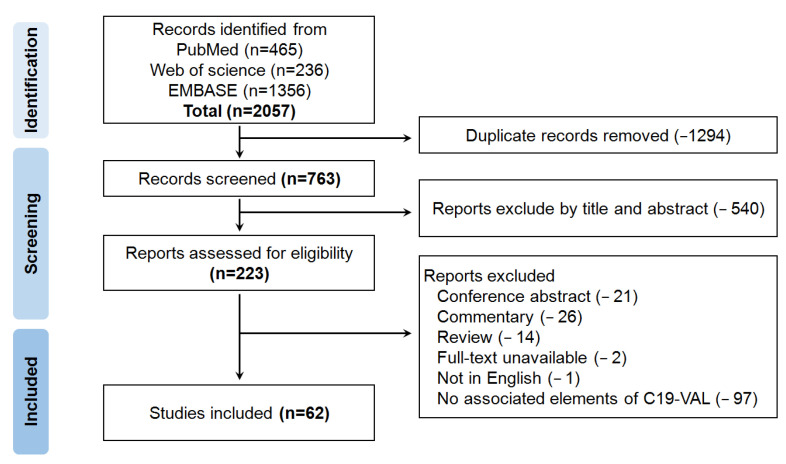
Flow chart of the systematic literature search and screening for studies of C19-VAL.

**Figure 2 vaccines-11-00696-f002:**
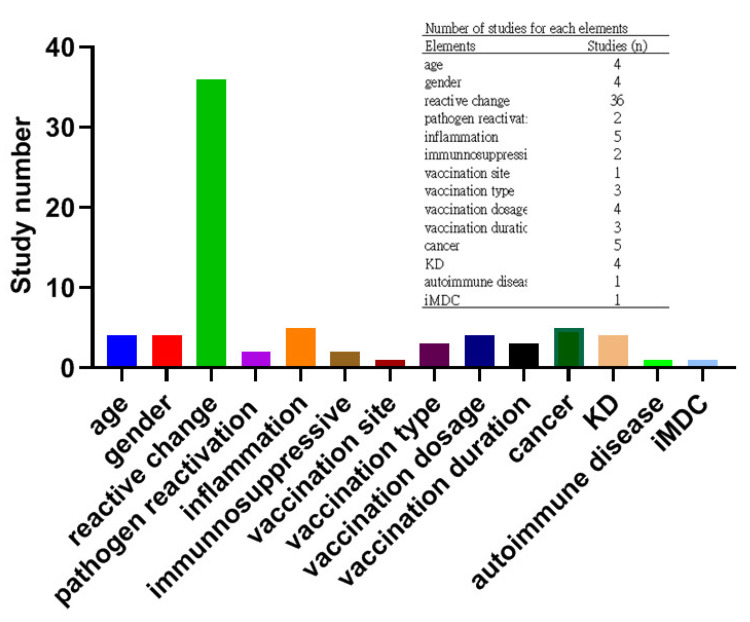
Study number regarding each element for C19-VAL. Other vaccine associated adverse effects were indicated as other adverse effects.

## Data Availability

Not applicable.

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
