# Peer review of "Immune Response Related to Lymphadenopathy Post COVID-19 Vaccination"

_vaccines, 2023, doi:10.3390/vaccines11030696_

Round 1

Reviewer 1 Report

The authors report information about the phenomenon of lymphadenopathy post COVID-19 vaccination in the context of its associated elements and mechanism.

The data is worth publishing, although the authors concede themselves that, although more than 60 studies were evaluated in detail, not many tangible conclusions can be drawn, due to either missing control groups, or a limited number of participants within a given study.

I recommend to have the revised manuscript checked by a native English speaker; in several instances, articles (a, the) are missing and incorrect use of adverbs or adjectives.

Specific comments:

Abstract:

Line 32, emphasize the characteristics of …

Line 50, in December 2020, with the goal to end in …

Line 58, 1.1 % are more than “rare”; in the German language, reporting of rare side-effects related to a medication would be 1 to 10 in 10000 individuals; the present 1.1 % would be classified as “occasional”; for the authors to elaborate on this point, please.

Line 73, please explain the abbreviation VLAD

Lines 80-82, what exactly is meant with “evaluate the associated elements”, evaluation in which respect?

Material and Methods

Line 88, the review focussed …

Results

Line 95, of the article

Figure 1, it seems that one number in the chart is incorrect: 530 reports were excluded and not 540, otherwise the downstream numbers would not be correct anymore, for the authors to verify, please.

Lines 105-107, 17 countries: USA is missing in the text with 15 studies; according to Table 1, Israel contributed with only 1 study.

Line 113, mostly reported

Figure 2 and lines 128-129, form the figure it visually appears that vaccination site, type, dosage and duration elements account for more than the 8 studies mentioned in lines 128-129.

Line 149, I am not sure about the meaning of “duration of vaccination”; is this the time post vaccination for symptoms of VAL to manifest?  Or the number of repetitive vaccinations before such symptoms elicit?

Line 152, 62 studies minus the 8 described in lines 126-150 is 54 studies, but 53 articles are mentioned here, for clarification, please.

Line 154, Figure 2

Line 158, … characterisation shown in 14 of 36 studies … show …

Line 169, exploring

Line 170, the text speaks about 7 studies, but reference is only made to 4.

Discussion

Line 203, suggest

Lines 217-219, please rephrase sentence, it is difficult to understand.

Line 221, received

Line 227, elicit a prominent

Line 248, needs to be explored

Conclusions

Line 263, close follow-up; what is the exact meaning? How would this be done?

I would suggest to emphasize in one or two sentences that more studies with a better and/or more comprehensive study design will be needed to be able to draw more tangible conclusion than those that were drawn in the manuscript based on the studies currently available.

Author Response

The authors report information about the phenomenon of lymphadenopathy post COVID-19 vaccination in the context of its associated elements and mechanism.

The data is worth publishing, although the authors concede themselves that, although more than 60 studies were evaluated in detail, not many tangible conclusions can be drawn, due to either missing control groups, or a limited number of participants within a given study.

I recommend to have the revised manuscript checked by a native English speaker; in several instances, articles (a, the) are missing and incorrect use of adverbs or adjectives.

Reply: The authors appreciated that the reviewer pointed out this comment. Our revised manuscript was checked by a native English speaker already.

Specific comments:

Abstract:

Line 32, emphasize the characteristics of …

Line 50, in December 2020, with the goal to end in …

Material and Methods

Line 88, the review focussed …

Results

Line 95, of the article

Line 113, mostly reported

Line 154, Figure 2

Line 158, … characterisation shown in 14 of 36 studies … show …

Line 169, exploring

Line 203, suggest

Line 221, received

Line 227, elicit a prominent

Line 248, needs to be explored

Reply: The authors appreciated that the reviewer pointed out those comments. All the suggestions above were revised and marked as red in the revised manuscript.

Line 58, 1.1 % are more than “rare”; in the German language, reporting of rare side-effects related to a medication would be 1 to 10 in 10000 individuals; the present 1.1 % would be classified as “occasional”; for the authors to elaborate on this point, please.

Reply: The authors appreciated that the reviewer pointed out this comment. To avoid confusion, it was revised as below.

In the clinical trial of COVID-19 vaccination, painful or nonpainful swelling or enlargement of the lymph node is an occasional side effect of vaccination (≤1.1% of incidence).

Line 73, please explain the abbreviation VLAD

Reply: The authors appreciated that the reviewer pointed out this comment. This typing error has been corrected as "VAL" in the revised manuscript.

Lines 80-82, what exactly is meant with “evaluate the associated elements”, evaluation in which respect?

Reply: The authors appreciated that the reviewer pointed out this comment. To avoid confusion, we changed the "evaluate" to "analyze".  

Figure 1, it seems that one number in the chart is incorrect: 530 reports were excluded and not 540, otherwise the downstream numbers would not be correct anymore, for the authors to verify, please. 

Reply: The authors appreciated that the reviewer pointed out this comment. However, there may be a misunderstanding. The 540 reports are excluded from the stage two screening (n=763), not from the initial screening "duplicate records removed (n=1294)".

Lines 105-107, 17 countries: USA is missing in the text with 15 studies; according to Table 1, Israel contributed with only 1 study.

Reply: The authors appreciated that the reviewer pointed out this comment. It has been corrected in the revised manuscript.

Figure 2 and lines 128-129, form the figure it visually appears that vaccination site, type, dosage and duration elements account for more than the 8 studies mentioned in lines 128-129.

Reply: The authors appreciated that the reviewer pointed out this comment. The site, type, dosage, and duration of vaccination for C19-VAL incidence was individually evaluated in the eight reports, which included six single-center retrospective studies, one case series and one case report (Table 1, [10, 13, 18-20, 42-44]). The vaccination site was obtained from ref 19; the vaccination type was obtained from ref 10, 13, 20; the vaccination dosage was obtained from ref 18, 20, 42, 43; the vaccination duration was obtained from ref 20, 43, 44. Because each reference may contain more than one element, the sum of those four elements is over eight.

Line 149, I am not sure about the meaning of “duration of vaccination”; is this the time post vaccination for symptoms of VAL to manifest?  Or the number of repetitive vaccinations before such symptoms elicit?

Reply: The authors appreciated that the reviewer pointed out this comment. It has been revised as " There is a negative correlation between the days of post-vaccination and C19-VAL incidence.".

Line 152, 62 studies minus the 8 described in lines 126-150 is 54 studies, but 53 articles are mentioned here, for clarification, please.

Reply: The authors appreciated that the reviewer pointed out this comment. However, there may be a misunderstanding. Among these 62 reports listed in Table 1, 53 articles which described the roles of age, gender, reactive change, inflammation, KD, and immunosuppression in C19-VAL (Table 1). These eight references described in lines of 126-150 were not excluded from the 62 reports. Also, those 53 papers were screened again from these 62 references (New Figure 2 in the revised manuscript).

Line 170, the text speaks about 7 studies, but reference is only made to 4.

Reply: The authors appreciated that the reviewer pointed out this comment. It has been corrected as below.

There were five studies exploring the role of cancer in the C19-VAL incidence [43, 84-87].

Discussion

Lines 217-219, please rephrase sentence, it is difficult to understand.

Reply: The authors appreciated that the reviewer pointed out this comment. It has been revised as below.

The lymphadenopathy immediately occurs in the LN after reactive hyperplasia (rapid extension) of T and B cells [96], which is a possible mechanism of C19-VAL. It also explains that the days of post-vaccination is a negative effect of C19-VAL incidence due to the reduction of reactive hyperplasia following the days of post-vaccination.

Conclusions

Line 263, close follow-up; what is the exact meaning? How would this be done?

I would suggest to emphasize in one or two sentences that more studies with a better and/or more comprehensive study design will be needed to be able to draw more tangible conclusion than those that were drawn in the manuscript based on the studies currently available. 

Reply: The authors appreciated that the reviewer pointed out this comment.

The conclusions were re-written as below.

C19-VAL has caused a dilemma in diagnosis and patient management, especially in oncological patients [8, 21]. It may lead to unnecessary biopsies and changes in therapy [8, 108], which increase the psychological and medical burden of patients and the risk of those having inherent diseases. To avoid the above issues, current recommendation suggests that screening exams can be scheduled before the first dose or 4-12 weeks after the second dose of COVID-19 vaccine [11]. For a boost dose, the exam should be provided at least 6 weeks post vaccination [108, 109]. Our results currently show that C19-VAL may be caused by a strong vaccine-induced immune response. In the future, additional studies are needed with a better and more comprehensive study design to draw more tangible conclusions.

With mass COVID-19 vaccination, the frequency of C19-VAL is rapidly increasing; more patients and clinicians will observe medical dilemmas from C19-VAL. However, the vaccine benefits exceed the medical dilemmas of C19-VAL. Therefore, we suggest a follow-up exam of C19-VAL patients as a precaution to resolve the dilemma in imaging diagnosis and patient management. Since the resolved time of C19-VAL is two months [13, 20], the follow-up exam should be processed by imaging for the size and number of C19-VAL within the two months of onset. If the C19-VAL is not resolved within this period, a biopsy examination should be suggested.

Reviewer 2 Report

Ho et al. conducted an extensive literature survey on the immune response related to lymphadenopathy post COVID-19 2 vaccination. They found several elements including receiver age and gender, reactive change within lymph nodes (LN), could have impact on the occurrence of C19-VAL. This review gave a preliminary insight of clinical finding regarding associated elements of C19-VAL, which will appeal extensive attentions from global researchers.

However, I will recommend it for publish after minor revisions.

1. In the Abstract, it is said this review ‘provide the mechanism of C19-VAL’. However, there was lack of the mechanism of C19-VAL in the main text.

2. P2L57: ‘In the clinical trial of COVID-19 vaccination, painful or nonpainful swelling or enlargement of the lymph node is a rare side effect of vaccination (≤1.1% of incidence).’ P2L63: ‘C19-VAL has become a common side effect..’ These two sentences made me feel confused whether C19-VAL is a rare or common side effect.

3. P2L57: ‘Types of articles contained single center retrospective study (n=10), cases series (n=6) and case report (n=47).’ The total numbers of articles mentioned in this sentence was 63, not 62.

4. The styles of references were not consistent with that suggested in the journal’s guidelines.

Author Response

Ho et al. conducted an extensive literature survey on the immune response related to lymphadenopathy post COVID-19 2 vaccination. They found several elements including receiver age and gender, reactive change within lymph nodes (LN), could have impact on the occurrence of C19-VAL. This review gave a preliminary insight of clinical finding regarding associated elements of C19-VAL, which will appeal extensive attentions from global researchers.

However, I will recommend it for publish after minor revisions.

  1. In the Abstract, it is said this review ‘provide the mechanism of C19-VAL’. However, there was lack of the mechanism of C19-VAL in the main text.

Reply: The authors appreciated that the reviewer pointed out this comment. The mechanism of C19-VAL may be induced by strong vaccine immune response through the B cell germinal center response post vaccination. We added some sentences at the end of abstract and in the section 3.3.

  1. P2L57: ‘In the clinical trial of COVID-19 vaccination, painful or nonpainful swelling or enlargement of the lymph node is a rare side effect of vaccination (≤1.1% of incidence).’ P2L63: ‘C19-VAL has become a common side effect..’ These two sentences made me feel confused whether C19-VAL is a rare or common side effect.

Reply: The authors appreciated that the reviewer pointed out this comment. To avoid confusion, it was revised as below.

In the clinical trial of COVID-19 vaccination, painful or nonpainful swelling or enlargement of the lymph node is an occasional side effect of vaccination (≤1.1% of incidence).

We also removed this sentence " C19-VAL has become a common side effect."

  1. P2L57: ‘Types of articles contained single center retrospective study (n=10), cases series (n=6) and case report (n=47).’ The total numbers of articles mentioned in this sentence was 63, not 62.

Reply: The authors appreciated that the reviewer pointed out this comment. It was a typo, the number of case report is 46. We corrected it. 

  1. The styles of references were not consistent with that suggested in the journal’s guidelines.

Reply: The authors appreciated that the reviewer pointed out this comment. The styles of references were modified according to the journal’s guidelines.

Reviewer 3 Report

The study of the side effects due to vaccine-induced lymphadenopathy has been widely reported over the past two years. A vast number of such systematic reviews and meta-analysis have been published. The current manuscript presents very limited new  results and no new knowledge that could inform decision making. 

It doesn't make sense to continue referencing existing knowledge in the conclusion section. The reviewer can't find any new insights from this manuscript. 

Author Response

The study of the side effects due to vaccine-induced lymphadenopathy has been widely reported over the past two years. A vast number of such systematic reviews and meta-analysis have been published. The current manuscript presents very limited new results and no new knowledge that could inform decision making. 

It doesn't make sense to continue referencing existing knowledge in the conclusion section. The reviewer can't find any new insights from this manuscript. 

Reply: The authors appreciated that the reviewer pointed out this comment.

The object of this study is to evaluate the associated elements of C19-VAL and to provide the mechanism of C19-VAL. The lymphadenopathy immediately occurs in the LN after reactive hyperplasia (rapid extension) of T and B cells, which is a possible mechanism of C19-VAL. It also explains that the days of post-vaccination is a negative effect of C19-VAL incidence due to the reduction of reactive hyperplasia following the days of post-vaccination. Our results currently show that C19-VAL may be caused by a strong vaccine-induced immune response. With mass COVID-19 vaccination, the frequency of C19-VAL is rapidly increasing; more patients and clinicians will observe medical dilemmas from C19-VAL. However, the vaccine benefits exceed the medical dilemmas of C19-VAL. Therefore, we suggest a follow-up exam of C19-VAL patients as a precaution to resolve the dilemma in imaging diagnosis and patient management. Since the resolved time of C19-VAL is two months, the follow-up exam should be processed by imaging for the size and number of C19-VAL within the two months of onset. If the C19-VAL is not resolved within this period, a biopsy examination should be suggested.

The sections of discussions and conclusions were revised.

Round 2

Reviewer 3 Report

The revised version made minor adaptions, but not addressed the core problem as raised in the previous round review. The reviewer maintains the criticism regarding the value of this piece of work to be published.  

The authors include an unreasonably large number of affiliations into the list, please check if it is appropriate.

Author Response

The comment of reviewer's suggestion was valuable. However, this manuscript type is a systematic review study, not a report of multiple cases. Our conclusions are based on the results of the literature review. Thus, we only summarize the points presented in the literature. We revised the abstract, and the conclusions were re-written again in the second revised manuscript. 

Due to work requirement, the corresponding author is co-employed in multiple affiliations. Although this is not related to the academic nature of this manuscript, we have removed some affiliations.